

# Sex expression and floral diversity in *Jatropha curcas*: a population study in its center of origin

María de Lourdes Adriano-Anaya, Edilma Pérez-Castillo, Miguel Salvador-Figueroa, Sonia Ruiz-González, Alfredo Vázquez-Ovando, Julieta Grajales-Conesa and Isidro Ovando-Medina

Instituto de Biociencias, Universidad Autónoma de Chiapas, Tapachula, Chiapas, Mexico

Corresponding author
Isidro Ovando-Medina,
isidro.ovando@unach.mx

## ABSTRACT

Sex expression and floral morphology studies are central to understand breeding behavior and to define the productive potential of plant genotypes. In particular, the new bioenergy crop *Jatropha curcas* L. has been classified as a monoecious species. Nonetheless, there is no information about its reproductive diversity in the Mesoamerican region, which is considered its center of origin and diversification. Thus, we determined sex expression and floral morphology in *J. curcas* populations from southern Mexico and Guatemala. Our results showed that most of *J. curcas* specimens had typical inflorescences with separate sexes (monoecious); meanwhile, the rest were atypical (gynoecious, androecious, andromonoecious, androgynomonoecious). The most important variables to group these populations, based on a discriminant analysis, were: male flower diameter, female petal length and male nectary length. From southern Mexico "Guerrero" was the most diverse population, and "Centro" had the highest variability among the populations from Chiapas. A cluster analysis showed that the accessions from southern Mexico were grouped without showing any correlation with the geographical origin, while those accessions with atypical sexuality were grouped together. To answer the question of how informative are floral morphological traits compared to molecular markers, we perform a Mantel correlation test between the distance matrix generated in this study and the genetic distance matrix (AFLP) previously reported for the same accessions. We found significant correlation between data at the level of accessions. Our results contribute to design genetic improvement programs by using sexually and morphologically contrasting plants from the center of origin.

## INTRODUCTION

*Jatropha curcas* L. is a plant that has recently attracted interest as a scientific model and as an agro-industrial crop due to oil content of its seeds. However, knowledge of this plant's biology and ecology is still limited, and recent studies locate the Mesoamerican region as its center of origin and diversity (*Montes-Osorio et al., 2014*).

Nonetheless, many studies about *J. curcas* have been performed with Asian and African accessions, where low genetic diversity is registered. Moreover, in the Mesoamerican
region, particularly in Chiapas, Mexico there are few studies on genetic variation. *Sánchez-Gutiérrez (2010)* studied 147 accessions from five populations (Istmo, Frontera, Frailesca, Centro, and Soconusco) using AFLP markers; she found that the largest variation was within populations (94.2%), while among populations the variation was 3.9%, and the "Istmo" population was the most diverse. In other study, *Pecina-Quintero et al. (2011)* analyzed 88 accessions from seven regions in Chiapas by using AFLP markers; they found that one of the accessions (Tuxtla Chico) provided 100% of pistillate flowers (female) and showed the highest number of rare fragments. Recently, *Montes-Osorio et al. (2014)* analyzed the relationship between morphological traits and AFLP markers in populations from Central America compared to Africa, Asia and South America. Mesoamerican accessions registered the highest phenotypic and molecular variation. These results are being used to identify QTL markers that contribute to improve agronomic performance for seed and oil productivity (*King et al., 2015*). Thus, the use of molecular markers and morphological traits may be useful to differentiate populations. *Heller (1996)* described *J. curcas* morphological traits, and found size changes in canopy, stem, root, bark and leaves. These variations on morphological traits are being observed in Indian landraces, on seed allometry (*Ginwal et al., 2005*) and morpho-physiological variation (*Saikia et al., 2009*). The parameters evaluated by *Saikia et al. (2009)* were plant height, stem girth, branches per plant and 100 seed weight. Nonetheless, there is a lack of information about inflorescences and flowers traits, which are considered highly conserved and could be used as estimators of *J. curcas* genetic diversity in Mesoamerica. In Chinese accessions (*Wu et al., 2011*), floral phenology was divided into twelve phases, where sexual differentiation in male and female flowers occurs in the seventh phase, besides that some plants possess mainly male and female flowers. To our knowledge, no sexual types and floral traits research exists in the center of origin. Therefore, this study aimed (a) to describe *J. curcas* sex diversity; (b) to assess the variability in Mesoamerican accessions with floral markers; and (c) to analyze the relationship between floral traits and molecular markers.

## MATERIALS AND METHODS

### Biological material and study site

We studied 103 *J. curcas* accessions (Table S1), collected in 33 sites in southern México (*Ovando-Medina et al., 2011a*). Three years old plants were grown without any agronomic management in *Jatropha* Germplasm Bank of the Universidad Autónoma de Chiapas (Mexico) (14.4976 N, 92.4774 W and 58 m above sea level; average annual temperature of 31 °C, average annual humidity 80%, average of 2,600 mm of rainfall and soil type andosol). From each accession, male flowers ($n = 40$) and female flowers ($n = 20$) were collected. Flowers were transported to the laboratory in polybags and stored at 4 °C for up to 48 h.

### Classification of flowers

The number of inflorescences was determined every 28 days during a year. Flowers were classified as female, male and hermaphrodite; first the pedicel was removed from each of the flowers and then photomicrographs were taken by using a compound microscope

(Zeiss Axiolab®) and/or stereoscope (Zeiss Stemi 2000 C®) equipped with a video camera (AxioCam MRc®) coupled to the Axio Vision © program.

## Floral traits

For all the flowers we determined the number of sepals, petals and nectaries. For male and hermaphrodite flowers the number of filaments, anthers and pollen count was determined. For female and hermaphrodite flowers the number of ovules and the size of the pistil and ovary were determined. We also measured the length and/or width and/or diameter and/or thickness of floral character. The presence of trichomes was evaluated according to the following scale: glabrous, moderately pubescent and abundantly pubescent flowers.

## Statistical analysis

In order to minimize differences observed in traits, our data were coded as shown in Table S2. These data were processed by a multivariate discriminant analysis, where the most informative characters were identified and populations were grouped. Then, a cluster analysis (Euclidean distance and Ward grouping method) was performed to visualize the relationships among populations. In both analyses two approaches were followed: one with all populations and the second one only with Chiapas populations (the most numerous). All statistical analyses were performed with the XLStat© v 2014 and InfoStat© v 2014 software. In addition, to answer the question of how informative are floral morphological traits compared to molecular markers, a Mantel correlation test was performed using GenAlEx© version 6.3 program. This correlation was carried out between data reported by *Sánchez-Gutiérrez (2010)* and the matrix of Fisher's distances obtained in this discriminant analysis. This procedure was done with five populations and 87 accessions.

# RESULTS AND DISCUSSION

## Flowering dynamics

In our results, flowers were observed throughout the year only in four accessions (ARR-7, CDCU-3, MAP-8 and PC-15). The rest blossomed from one to three times a year; 85.2% flowered from March to April, while in 59% flowered during August and only 40.1% of the accessions flowered at the start of November (Fig. S1).

Many factors affect the start of flowering, like genetic variability, nutrients, phyto-hormones and environmental conditions. Soil humidity may be involved in triggering flower formation because we observed a clear relationship between the beginning of rainy season and *J. curcas* flowering (Fig. S2). Control of soil humidity could be key to induce continuous blooming, as suggested by *Sukarin, Yamada & Sakaguchi (1987)*. Geographic location influences the period, intensity and frequency of rain, thus flowering peak changes, e.g., in Nicaragua this was observed during April, May, June and August (*Aker, 2012*) and in India from July to September (*Sukarin, Yamada & Sakaguchi, 1987*; *Bhattacharya & Kumar, 2005*).

*J. curcas* has terminal inflorescences (*Halle, Oldeman & Tomlinson, 1978*), so the number of them in the plant depends on the number of mature branches (terminal sprouts) present at the time of sampling. From our results, the genetic determinant of flowering in *J.*

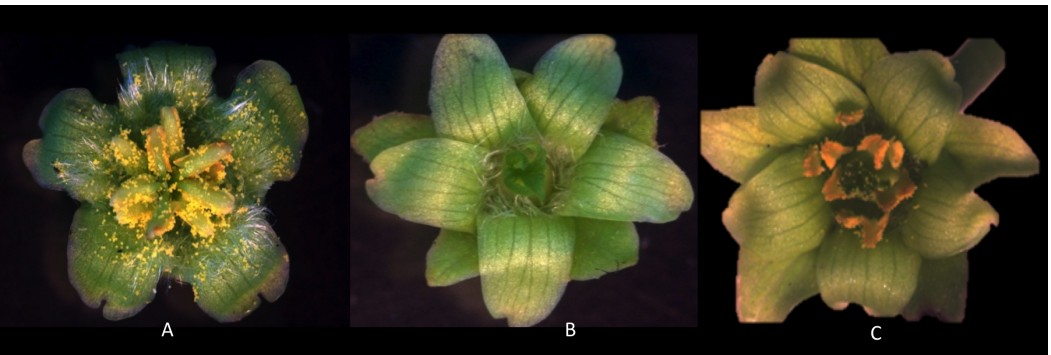

**Figure 1** **Types of *Jatropha curcas* flowers.** Differentiation of *Jatropha curcas* flowers according to sex, male flower (A), female flower (B), hermaphroditic flower (C).

**Table 1** **Sexuality of *Jatropha curcas* landraces collected in Meso-America.** Classification of 103 accessions of *Jatropha curcas* L. from the *Jatropha* Germplasm Bank of the Universidad Autónoma de Chiapas, based on their sexual descriptors.

| Type of plant | Accessions (*n*) | Description |
|---|---|---|
| Monoecious | 95 | Plants with male and females flowers in the same inflorescence |
| Dioecious | | |
|     Gynoecious | 3 | Plants with only female flowers |
|     Androecious | 2 | Plants with only male flowers |
| Hermaphrodites | | |
|     Andromonoecious | 1 | Plants with male and hermaphrodite flowers |
|     Androgynomonoecious | 2 | Plants that has mostly male flowers, with a few female and hermaphrodite flowers |

*curcas* accessions used in this study seems a highly influential factor, since the number of inflorescences per plant was accession-specific.

## Sex of flowers in *J. curcas* accessions

We found female, male and hermaphrodite flowers in the study accessions (Fig. 1), and based on the proportion of each flower sex, plants were classified as gynoecious, androecious, andromonoecious, androgynomonoecious (Table 1). *Dehgan & Webster (1979)* reported that *J. curcas* is known as a monoecious plant (male and female flowers on the same inflorescence, also with hermaphrodite ones), without specifying if this is a genotype-specific characteristic. Some other studies reported plants with only female flowers (*Pecina-Quintero et al., 2011*) or predominantly with male flowers (*Wu et al., 2011*).

The presence of dioecism and complex arrangements, and the fact that most individuals were monoecious indicate that in this plant sex is linked to a complex determinant. Probably *J. curcas* has sex chromosomes like to other species with similar sex expression (*Charlesworth, 2016*). Dioecism, in plants, is related to various ecological factors (*Vamosi et al., 2009*) and is observed in several groups of unrelated plants. The appearance of dioecism has been reported in the genus *Ribers* (*Senters & Solis, 2003*) and has appeared
at least twice in the genus *Silene* from a gynodioecious ancestor (*Desfeux et al., 1996*). In particular, in the Siparunaceae family, dioecism was originated from a monoecious ancestor (*Renner & Won, 2001*). In monocots the dioecious nature seems to have evolved from a transient gynodioecious state more than the monoecious state (*Weiblen, Oyama & Donoghue, 2000*). The existence of complex systems has been documented in dioecious crops as apple (*Hancock, 2012*) and hemp (*Mandolino et al., 1999*).

In contrast to animals, the sexual dimorphism appearance in angiosperms was not associated with the evolution of sex chromosomes, with exception of a few species (*Ruiz, 2004*). Heteromorphic sex chromosomes are widely distributed in higher animals but in contrast to the study of the origin and evolution of X and Y chromosomes in humans (*Skaletsky et al., 2003*), plants sex chromosomes have been scarcely studied. It is estimated that animals sex chromosomes appeared 240–320 million years ago (*Charlesworth, 2002*). Therefore, younger systems like dicots (*Ruiz, 2004*) are interesting to prove this hypothesis. Although sexual determinism in plants and animals is fundamentally different, the architecture of sex chromosomes is likely to be comparable (*Charlesworth, 2002*). It has been hypothesized that sex chromosomes evolved from autosomes and have required at least two evolutionary events for the transition from hermaphroditism to dioecism (*Charlesworth, 1991*). Therefore, a male sterility mutation in hermaphrodites (gynomonoecious) would give way to females, and then the males would appear from the suppression of female genes. However, in this study with *J. curcas* collected in its center of origin and diversity (*Salvador-Figueroa et al., 2015*; *Pamidimarri & Reddy, 2014*) we did not find gynomonoecious individuals, which could be possible by (a) *J. curcas* gynoecious plants originated by one mutational step from a hermaphrodite ancestor, or (b) the gynodioecia emerged from androgynomonoecious populations (Fig. 2). Therefore, it is necessary to perform a study involving more individuals to confirm the absence of gynomonoecious. Reminiscences of sex chromosomes could be found in androgynomonoecious and andromonoecious individuals.

## Variation of *J. curcas* floral characters

We observed casual and recurrent variations in *J. curcas* floral traits. The first type refers to the variability among inflorescences of the same plant, which may have the origin in the genotype-environment interaction (*Heller, 1996*) or in epigenetic factors (*Yi et al., 2010*). Examples of occasional variation in male flowers are: number of petals (four or five), sepals (four, five or six), nectaries (four or five) and stamens (seven to ten). In female flowers occasional variation was found in the number of ovules (two, three or four). The second type of variation refers to features that were always present in the same plant, such as the amount of trichomes and the size of the characters. It is possible that the characters had recurrent variation, whether fixed or highly heritable (*Ovando-Medina et al., 2011a*). Only the recurrent variation was used for studies of diversity on *J. curcas* populations from southern México.

Our results revealed that the typical male flowers whorls (Fig. 1A) were composed of five petals, five sepals, five nectaries, ten stamens, ten anthers and pollen grains (27–44 μm diameter), and female flowers (Fig. 1B) have five petals, five sepals, five nectaries, one

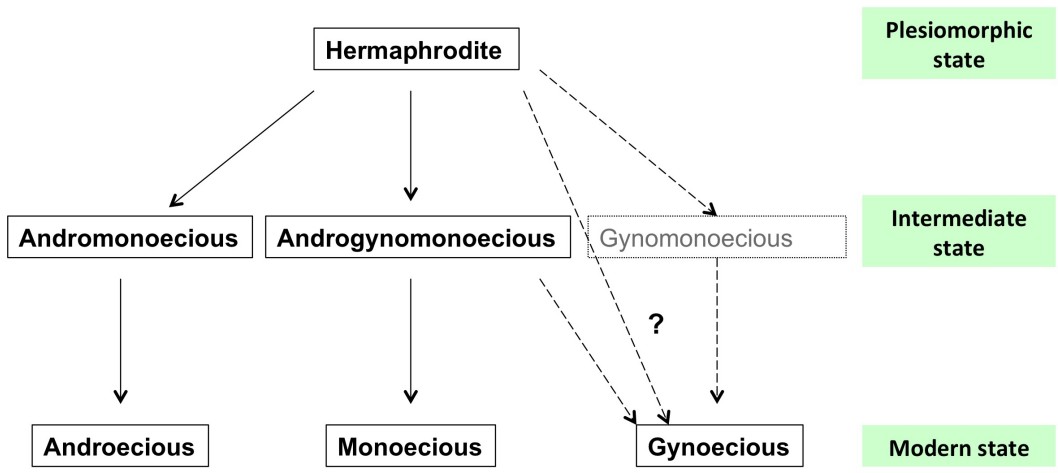

**Figure 2 Sex expression in the biofuel plant *Jatropha curcas*.** Phylogenetic hypothesis of the evolution of sexuality in the tropical tree *Jatropha curcas* L. (Euphorbiaceae).

ovary, three ovules, one pistil. In hermaphrodite flowers (Fig. 1C) there are five petals, five sepals, five nectaries, ten stamens, pollen grains ranging from 37–40 μm in diameter, one ovary, three ovules, one pistil. These data agree with those previously reported by other authors (*Heller, 1996*; *Toral et al., 2008*; *Wu et al., 2011*).

Regarding flowers size, we found that male flowers are smaller (5–10 mm) than female (7–15 mm) and hermaphrodites (11–14 mm) ones; they also have a light green-yellow color, with most of flowers moderately pubescent. In contrast, *Toral et al. (2008)* mentioned that male and female flowers are small (6–8 mm), have a greenish-yellow color, with some pubescent flowers, and with versatile anthers and protruding sexual organs.

We also observed two hermaphrodite flower types; one with ten stamens around the ovary longer than the pistil and the second one with four to six stamens around the ovary shorter than the pistil. The first type could be considered as a strategy to be self-fertilized, because the pollen sac opens, and this can be used by the receptive stigma, without needing a dispersing agent (*Ocampo-Velázquez, Malda-Barrera & Suárez-Ramos, 2009*). *Pinilla et al. (2011)* observed that *J. curcas* hermaphrodite flowers exhibit protrandy, since the development of the male and female phases first release pollen by manifesting as gynoecium growth in style and stigma, once produced senescence in the anthers, styles reach their ideal size when the papillae are already developed.

## Discriminant analysis of floral morphological characters

For the first discriminant analysis we used all populations from Southern Mexico, and the most important characters were identified to form the five principal components and to group accessions by populations. Principal component 1 ($C1_A$) was the most important accounting for 46.02% of the total variation (Table 2) and the most important variables for their contribution to $C1_A$ were: male flower diameter (MFD), female sepal length (FSL), female petal length (FPL), female flower diameter (FFD) and pistil thickness (PT) (Table S3). Principal component 2 ($C2_A$) accounted for 20.90% of the variance (Table 2)

**Table 2  Discriminant analysis of flower traits of *Jatropha curcas* collected in Meso-America.** Eigenvalues of the five main components, based on floral characters of 103 accessions of *Jatropha curcas* L. in southern Mexico.

| Component | Eigenvalues | Variance (%) | Cumulative variance% |
|---|---|---|---|
| 1 | 2.430 | 46.018 | 46.018 |
| 2 | 1.104 | 20.898 | 66.916 |
| 3 | 0.720 | 13.642 | 80.558 |
| 4 | 0.445 | 8.424 | 88.982 |
| 5 | 0.264 | 5.001 | 93.983 |

and the variables that contributed most to this factor were: pistil thickness (PT), ovary length (OL), male nectary length (MNL) and female nectary length (FNL). The variables correlated to the main component 3 (F3) were ovule length (OL), quantity of trichomes in female flowers (TF) (Table S3), with 13.64% of the variance, whereas the main component 4 and 5 only explained 5–8% of the total variation as shown in Table 2.

Many authors have performed studies on *J. curcas* morphological variation by using principal component analysis, e.g., *Zapico et al. (2011)* in the Philippines evaluated 21 quantitative morphological variables for 13 accessions of *J. curcas*, the five principal components accounted for 88.12% of the total variation, where the most important variables for the principal component 1 were: plant height, number of leaves, percentage of seed germination and survival. Moreover, *Vijayanand et al. (2009)*, studied 12 accessions of *J. curcas* to assess genetic diversity using 19 morphological characters, finding that the first three factors contributed 89.2% of the total observed variance, the variables that contributed most to the component 1 were: stem diameter, length and width of the leaf, and plant growth, which contributed 35.7%. *Machado (2011)* conducted a study to characterize the morphological and productive variability of a collection of *J. curcas* in Cuba, taking into account 13 morphological characters, finding that plant height was the most variable indicator, followed by the thickness of stem and primary branches, number of primary and secondary branches, accounting for 74.63% between the principal component 1 and principal component 2. However, there have been no previous investigations on floral variation found in *J. curcas* for possible comparison.

We observed that the grouping pattern of accessions by population, according to $C1_A$ and $C2_A$, accounted for 66.92% of the variance shown (Fig. 3), where accessions belonging to Frontera and Oaxaca populations were grouped into $C1_A$, while accessions belonging to the Istmo and Centro populations in the $C2_A$. Guerrero's position in the lower left quadrant is due to the high percentage of variance shown by these accessions in the evaluated characters and their contribution to the $C1_A$ and $C2_A$. These results are probably due to existing plants in this population with hermaphrodite flowers (see arrows in Fig. 3).

The relationship between populations and centroids were plotted in Fig. 4, in order to show how Centro and Istmo populations are closely related since Istmo is found within the centroid of Centro population. Also, this population is related to the populations of Oaxaca and Frailesca, which may be due to the geographical proximity between them.

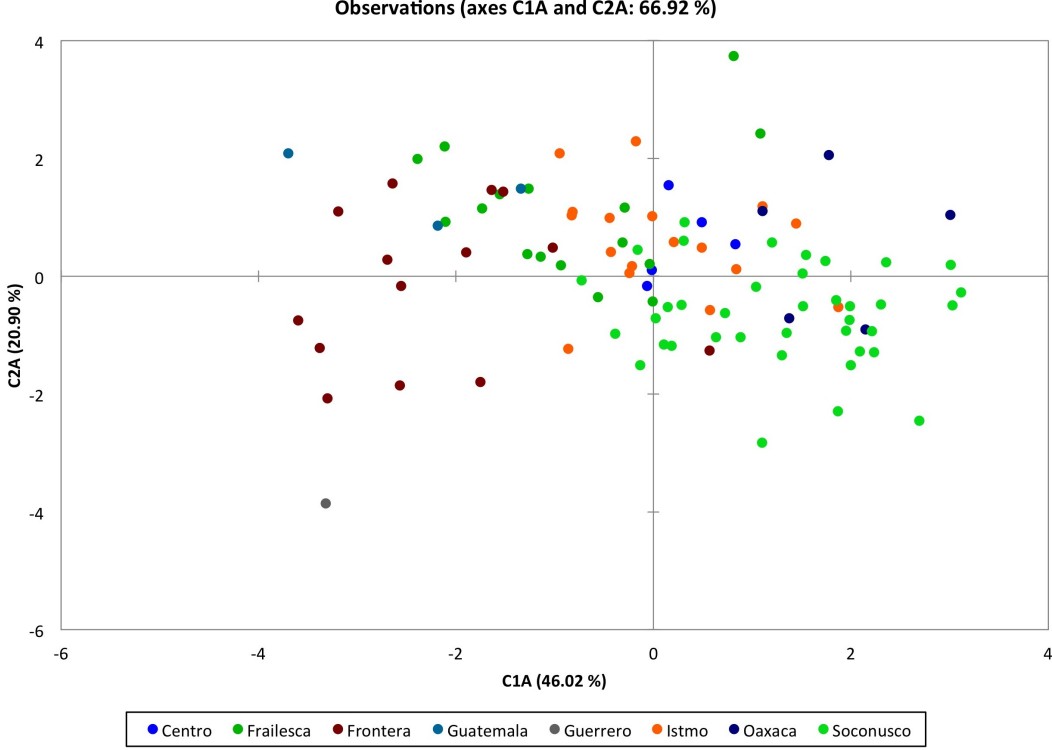

**Figure 3 Discriminant analysis of landraces of *Jatropha curcas* from Meso-America.** Groupings by populations of 103 accessions of *Jatropha curcas* L. of southern México, from the *Jatropha* Germplasm Bank of the Autonomous University of Chiapas. Traits: Male flower diameter, Male sepal length, Male sepal width, Male petal length, Male petal width, Male nectary length, Male nectary thickness, Filament length, Filament thickness, Anther length, Anther thickness, Pollen diameter, Trichomes in male flowers, Female/hermaphrodite flower diameter, Female/hermaphrodite sepal length, Female/hermaphrodite sepal width, Female/hermaphrodite petal length, Female/hermaphrodite petale width, Female/hermaphrodite nectary length Female/hermaphrodite nectary thickness, Pistil length, Pistil thickness, Ovary length, Ovary thickness, Ovule length, Ovule thickness, Trichomes in female/hermaphrodite flowers.

Meanwhile, populations of Guatemala, Frontera, Soconusco and Guerrero were separated from the rest and between themselves. It is remarkable how different are the Frontera and Soconusco populations, even though they are geographically closed, this may be due to the physical barrier represented by the Sierra Madre de Chiapas. Moreover, Guerrero was the most diverse population, probably because there are plants with hermaphrodite flowers in this population.

*Ovando-Medina et al. (2011a)*, used the same accessions and studied the genetic diversity estimated with fatty acids of *J. curcas* seeds, they reported that the variation with respect to oil content was 8.020%–54.28%. They also found that the two principal components together explained 89.25% of the total variation. By a graph of centroids the relationship between the six populations was shown; the results were similar to the observed by grouping based on floral characters variation, as Guatemala was one of the most diverse populations in both studies, in addition to the close relationship in which the populations of the center and coast of Chiapas were grouped.

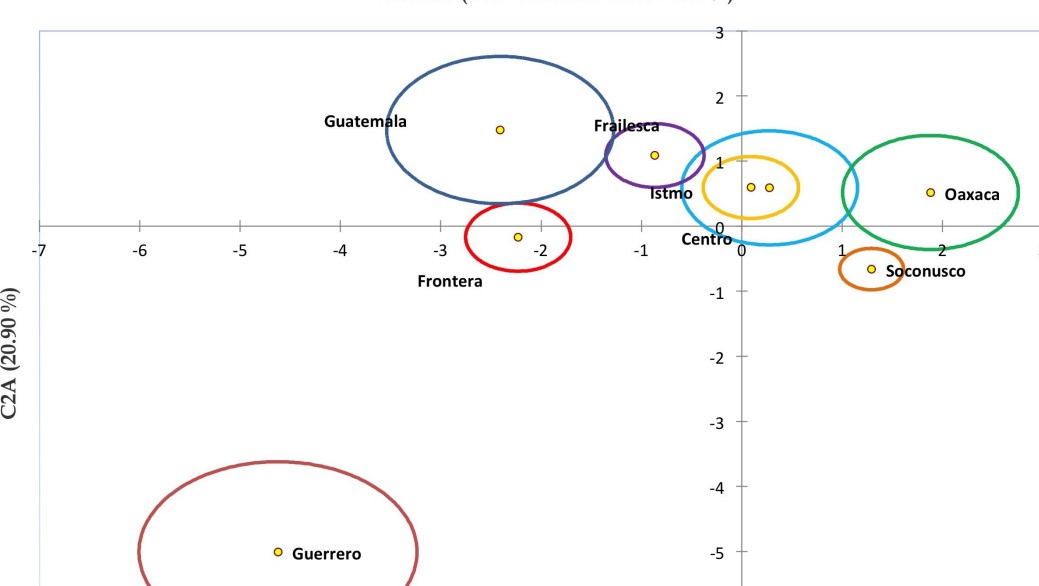

**Figure 4  Grouping of *Jatropha curcas* landraces based on discriminant analysis.** Graph of centroids of the eight populations of *Jatropha curcas* L. in southern México, from the *Jatropha* Germplasm Bank of the Autonomous University of Chiapas. Traits: Male flower diameter, Male sepal length, Male sepal width, Male petal length, Male petal width, Male nectary length, Male nectary thickness, Filament length, Filament thickness, Anther length, Anther thickness, Pollen diameter, Trichomes in male flowers, Female/hermaphrodite flower diameter, Female/hermaphrodite sepal length, Female/hermaphrodite sepal width, Female/hermaphrodite petal length, Female/hermaphrodite petale width, Female/hermaphrodite nectary length Female/hermaphrodite nectary thickness, Pistil length, Pistil thickness, Ovary length, Ovary thickness, Ovule length, Ovule thickness, Trichomes in female/hermaphrodite flowers.

We also carried out a second discriminant analysis, which exclusively studied accessions from Chiapas, because in that State there is the greatest extension of traditional cultivation of *J. curcas* in Mexico. The major characters were identified to form the five principal components and grouped the accessions by populations. The most important component (C1$_B$) explained 58.40% of the total variation (Table 3) and its most informative variables were diameter of the male flower (DMF), length of male sepal (LMS), male petal width (APM) and female petal length (LPF). Principal component 2 (C2$_B$) accounted for 20.92% of the variance (Table 3) and the most important variables were length of male nectary (LNM), width of the male anther (AAM), length of the ovule (LOvu) and the amount of trichomes on female flowers (TF). For C3$_B$ the most important variables were: the number of trichomes on male flowers (TM) and length of ovary (LO) accounting for a 16.81% variance, while the C4$_B$ contributed less variation than 4% of the total variance, as shown in Table 3. Contributions of studied traits to construct the principal components are showed in Table S4.

**Table 3** **Discriminant analysis of flower traits of *Jatropha curcas* from Meso-America.** Eigenvalues of the four principal components, based on floral characters of 90 accessions of *Jatropha curcas* L. in the State of Chiapas, Mexico.

| Component | Eigenvalues | Variance (%) | Cumulative variance (%) |
|---|---|---|---|
| 1 | 1.984 | 58.40 | 58.40 |
| 2 | 0.711 | 20.92 | 79.320 |
| 3 | 0.571 | 16.81 | 96.14 |
| 4 | 0.131 | 3.87 | 100 |

Considering the two discriminant analyses, the most important variables for all populations were the diameter of the male flower (DFM), female petal length (LPF), length of male nectary, male flower diameter (DFM), female petal length (LPF) and length of male nectary (LNM).

*Pinilla et al. (2011)* studied 246 accessions of *J. curcas* from Colombia through the study of 24 qualitative and quantitative morphological characters, finding that plant height, stem diameter, canopy projection, length and width of the eighth leaf, days flowering, weight and fruit length, length and width of the seed, explained a 93.62% of the variance.

In Fig. 5, the grouping pattern of accessions per population, according to $C1_B$ and $C2_B$, which together account for 79.31% of the variance, shows how accessions belonging to the population Soconusco is the only group around $C1_B$, while populations of Istmo and Centro are grouped at $C2_B$.

The relationships between the five populations may be visualized when the centroids (Fig. 6) are plotted, where we observed a group containing the Istmo, Frailesca and Centro populations. Frontera and Soconusco populations showed a marked differentiation between them and with respect to other populations. This shows that the Sierra Madre de Chiapas is a strong physical barrier between these populations and may be the main cause of diversification among populations. Centro was the most diverse population.

## Cluster analysis of morphological characters of flowers

Two analyses of hierarchical ascendant classification (dendrogram or cluster analysis) were undertaken: one of them analyzed 103 accessions from southern México, finding a dendrogram with five groups as shown in Fig. 7. The first group was the largest and contained 54 accessions from most populations, showing no correlation with geographical origin. The second group only gathered eight accessions, the majority from the population of Istmo. The third group consists of 35 accessions, which belongs to Centro, Frontera, Soconusco, Guatemala and Veracruz populations. The fourth group consists of ginoecious plants i.e., that only produce female flowers. The last group was formed from two androecious accessions (plants), i.e., that only produce male flowers, and one andromonoecious plant, i.e., producing hermaphrodite and male flowers. There was no group according to geographical origin. The percentage of variation within groups was 59.43%, while between groups was 40.57%, which means that larger variance is within groups.

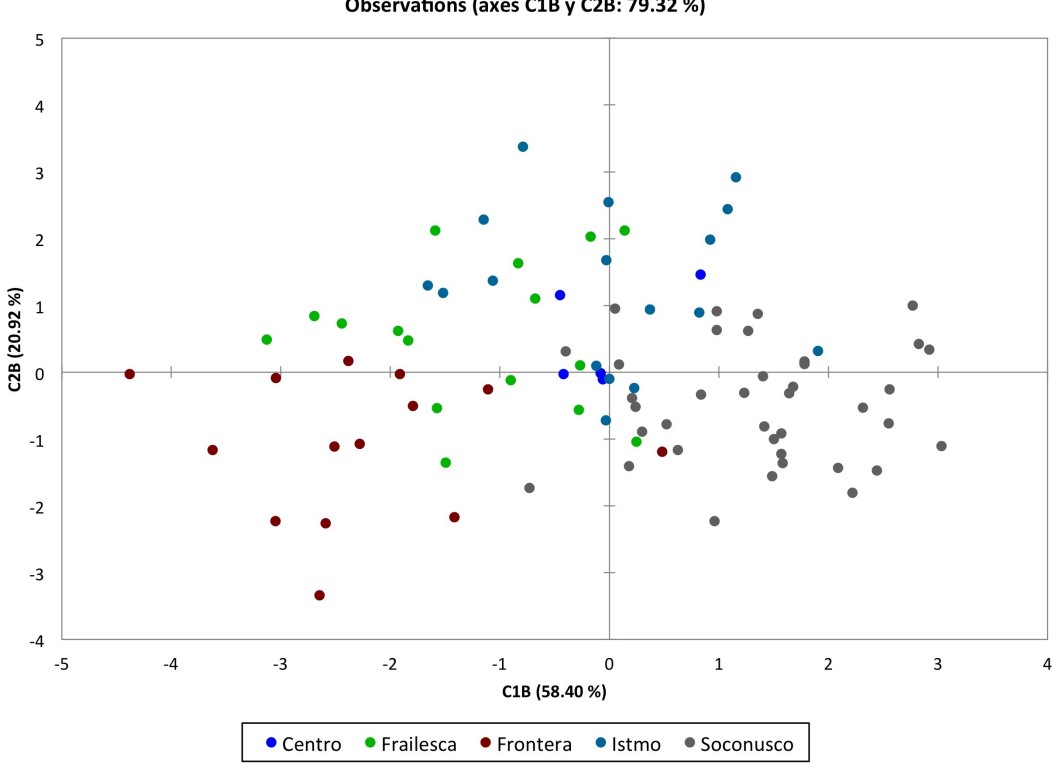

**Figure 5** **Discriminant analysis of *Jatropha curcas* landraces based on floral traits.** Grouping by populations of 90 accessiones of *Jatropha curcas* L. from Chiapas state, from the *Jatropha* Germplasm Bank of the Autonomous University of Chiapas. Traits: Male flower diameter, Male sepal length, Male sepal width, Male petal length, Male petal width, Male nectary length, Male nectary thickness, Filament length, Filament thickness, Anther length, Anther thickness, Pollen diameter, Trichomes in male flowers, Female/hermaphrodite flower diameter, Female/hermaphrodite sepal length, Female/hermaphrodite sepal width, Female/hermaphrodite petal length, Female/hermaphrodite petale width, Female/hermaphrodite nectary length Female/hermaphrodite nectary thickness, Pistil length, Pistil thickness, Ovary length, Ovary thickness, Ovule length, Ovule thickness, Trichomes in female/hermaphrodite flowers.

In the second analysis of hierarchical ascendant classification (dendrogram or cluster analysis) 90 accessions from the Chiapas state were studied and our results showed five groups (Fig. 8). The first group was the second largest consisting of 38 accessions, also from all populations (Soconusco, Frontera, Istmo and Centro), but most were from Soconusco populations, while in the second group eight accessions were included, most from the Istmo population. The third group was the largest and consisted of 39 accessions of all populations. The fourth group was formed by two accessions; these plants were androecious, i.e., only produce male flowers. Finally, the last group consists of three gynoecious plants, i.e., accessions with only female flowers. Groups showed no correlation according to their geographical origin. The percentage of variation within groups was 57.55%, while between groups was 42.45%, meaning that the existing variance is greatest within groups.

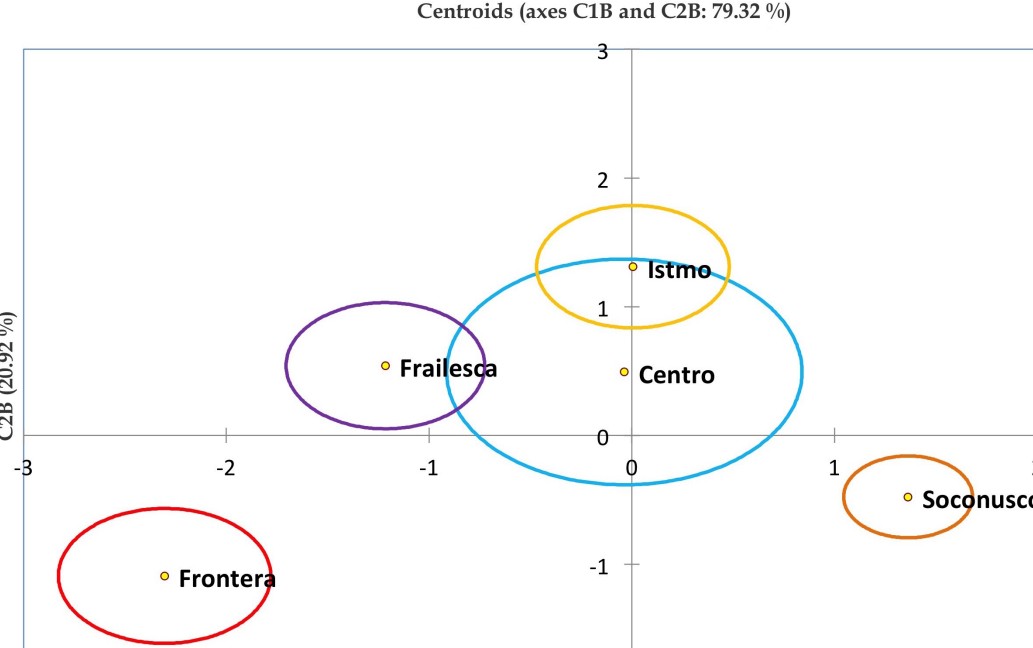

**Figure 6 Biplot of discriminant scores based on discriminant functions 1 and 2.** Chart of centroids of five populations of *Jatropha curcas* L. Chiapas state, from the *Jatropha* Germplasm Bank of the Autonomous University of Chiapas. Traits: Male flower diameter, Male sepal length, Male sepal width, Male petal length, Male petal width, Male nectary length, Male nectary thickness, Filament length, Filament thickness, Anther length, Anther thickness, Pollen diameter, Trichomes in male flowers, Female/hermaphrodite flower diameter, Female/hermaphrodite sepal length, Female/hermaphrodite sepal width, Female/hermaphrodite petal length, Female/hermaphrodite petale width, Female/hermaphrodite nectary length Female/hermaphrodite nectary thickness, Pistil length, Pistil thickness, Ovary length, Ovary thickness, Ovule length, Ovule thickness, Trichomes in female/hermaphrodite flowers.

The results coincide with those found by *Sánchez-Gutiérrez (2010)*, who used the same accessions to study genetic variation in *J. curcas* in the state of Chiapas. The author found using cluster analysis, that the accessions are grouped without a geographical pattern.

The Mantel test results showed that matrices generated by populations are not correlated ($p = 0.448$), while the matrices generated by accessions did show correlation ($p = 0.001$). This means that the study of the diversity of *J. curcas* using floral morphological markers reveals grouping patterns in accessions similar to those obtained with AFLP molecular markers.

Although there are many studies of *J. curcas* diversity collected in different regions of the world (for a review see *Ovando-Medina et al., 2011b*), no reports use floral markers as estimators of variation, so we were unable to compare the findings of this study. A characterization study of floral development in *J. curcas* was reported by *Wu et al. (2011)*, who discussed in detail the anatomy of flowers and inflorescences, but did not present data on variation between accessions.

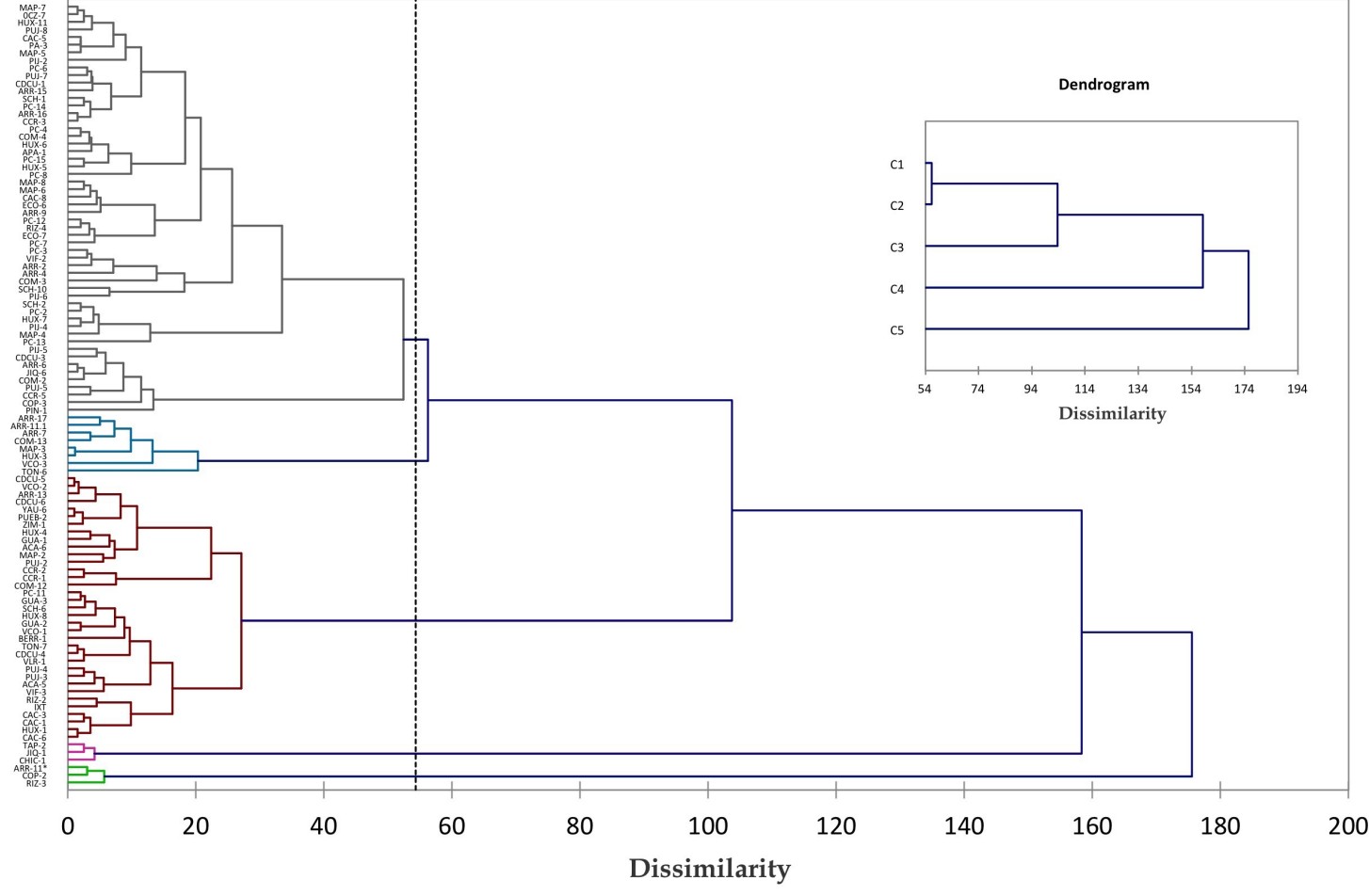

**Figure 7** **Clustering of *Jatropha curcas* accessions collected in Meso-America.** Dissimilarity dendrogram of 103 accessions of *Jatropha curcas* L. in southern México, from the *Jatropha* Germplasm Bank of the Autonomous University of Chiapas. Colors of branches denote groups formed. Traits: Male flower diameter, Male sepal length, Male sepal width, Male petal length, Male petal width, Male nectary length, Male nectary thickness, Filament length, Filament thickness, Anther length, Anther thickness, Pollen diameter, Trichomes in male flowers, Female/hermaphrodite flower diameter, Female/hermaphrodite sepal length, Female/hermaphrodite sepal width, Female/hermaphrodite petal length, Female/hermaphrodite petale width, Female/hermaphrodite nectary length Female/hermaphrodite nectary thickness, Pistil length, Pistil thickness, Ovary length, Ovary thickness, Ovule length, Ovule thickness, Trichomes in female/hermaphrodite flowers. For details of grouping mode see the Materials an Methods section.

Our results contribute to *J. curcas* sex expression knowledge from the living fences in southern México, which indicated, together with data from other research using phenotypic and molecular markers, that the Mesoamerican region is a center of diversification for this species.

## CONCLUSIONS

*Jatropha curcas* L. floral traits in the Mesoamerican region are highly variable between accessions. Our results showed that male flower diameter, female petal length, male nectary length are the most informative. Even though 93.2% of the accessions were monoecious, the rest of them were androecious, gynoecious, andromonoecious or androgynomonoecious.
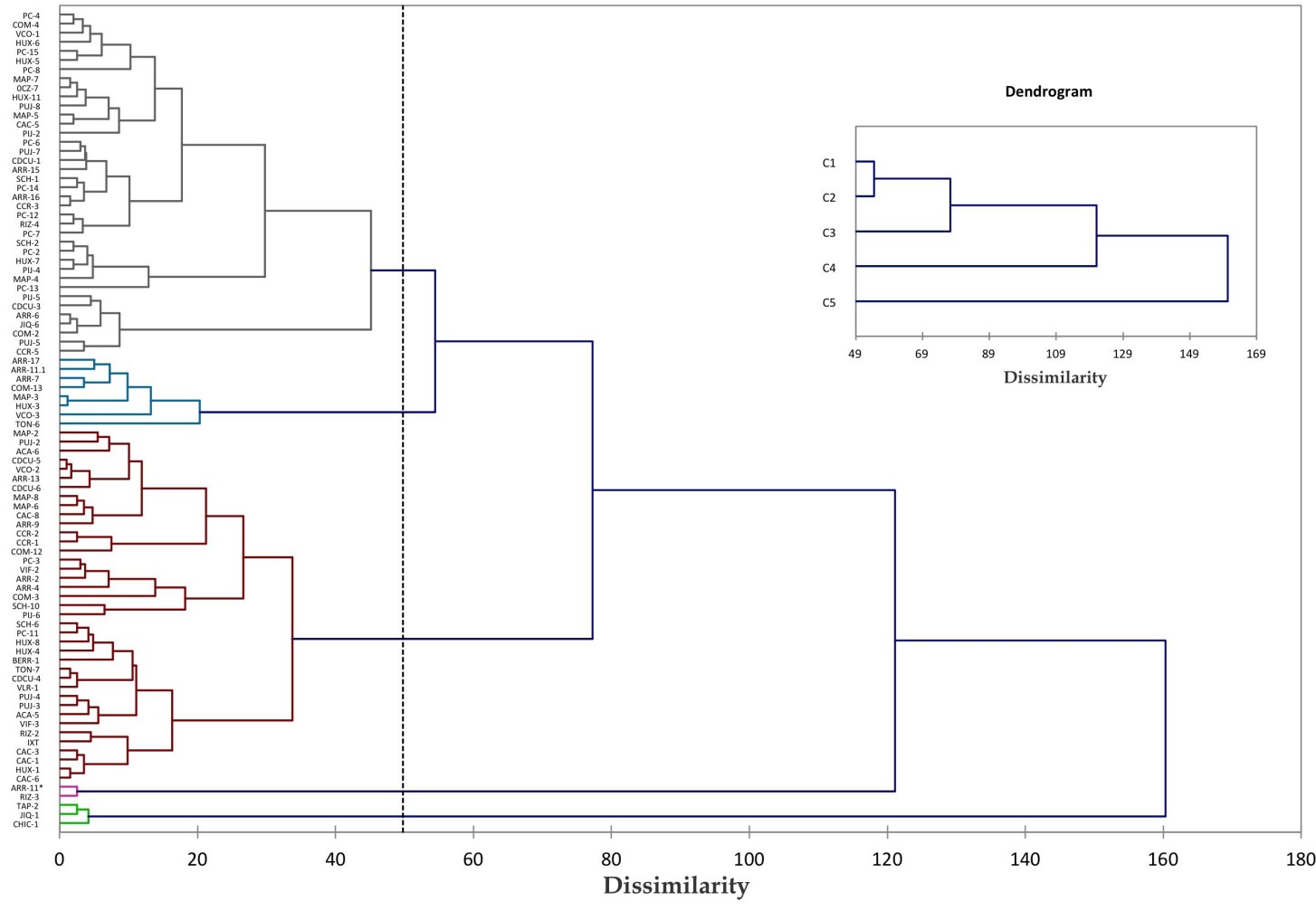

**Figure 8  Clustering of *Jatropha curcas* accessions collected in the State of Chiapas, Mexico.** Dissimilarity dendrogram of 90 accessions of *Jatropha curcas* L. of Chiapas state, from the *Jatropha* Germplasm Bank of the Autonomous University of Chiapas. Colors of branches denote groups formed. Traits: Male flower diameter, Male sepal length, Male sepal width, Male petal length, Male petal width, Male nectary length, Male nectary thickness, Filament length, Filament thickness, Anther length, Anther thickness, Pollen diameter, Trichomes in male flowers, Female/hermaphrodite flower diameter, Female/hermaphrodite sepal length, Female/hermaphrodite sepal width, Female/hermaphrodite petal length, Female/hermaphrodite petale width, Female/hermaphrodite nectary length Female/hermaphrodite nectary thickness, Pistil length, Pistil thickness, Ovary length, Ovary thickness, Ovule length, Ovule thickness, Trichomes in female/hermaphrodite flowers. For details of grouping mode see the Materials an Methods section.

Despite their geographical proximity, the Frontera and Soconusco populations are differentiated in their floral characters.

The multivariate analysis registered that *J. curcas* accessions from southern México were grouped without a geographic pattern, while accessions with atypical sexuality were grouped together; in this way, one group was formed by gynoecious plants and another by androecious and andromonoecious accessions.

### Funding

This work was supported by a research grant provided by the Agricultural Innovation MKTPlace Initiative (Regulation of the flowering of Jatropha curcas to improve the sustainability of biofuel feedstock production by farmers in Latin America and the Caribbean). The funders had no role in study design, data collection and analysis, decision to publish, or preparation of the manuscript.

### Grant Disclosures

The following grant information was disclosed by the authors:
The Agricultural Innovation MKTPlace Initiative.

### Competing Interests

The authors declare there are no competing interests.

### Author Contributions

- María de Lourdes Adriano-Anaya conceived and designed the experiments, analyzed the data, contributed reagents/materials/analysis tools.
- Edilma Pérez-Castillo performed the experiments, prepared figures and/or tables.
- Miguel Salvador-Figueroa conceived and designed the experiments, contributed reagents/materials/analysis tools, reviewed drafts of the paper.
- Sonia Ruiz-González performed the experiments, prepared figures and/or tables.
- Alfredo Vázquez-Ovando performed the experiments, analyzed the data, wrote the paper.
- Julieta Grajales-Conesa performed the experiments, wrote the paper.
- Isidro Ovando-Medina conceived and designed the experiments, analyzed the data, wrote the paper, reviewed drafts of the paper.

### Data Availability

   Raw data are provided as Supplemental Information.

### Supplemental Information

Supplemental information for this article can be found online at http://dx.doi.org/10.7717/peerj.2071#supplemental-information.

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
