# Peer review of "Sex expression and floral diversity in Jatropha curcas: a population study in its center of origin"

_PeerJ, doi:10.7717/peerj.2071_

## Round 0.1 · original submission · Major Revisions

The manuscript can be considered for publication after taking into account the suggestions of the reviewers. Please, note the comments in the attached version of the manuscript.

Some additional minor corrections:

line 63: omit ;

line 78: Jatropha

In line 263 table 5 is cited, which does not exist

Figures 7 and 8: explain colour coding within the dendrograms

·

Basic reporting

English text should be corrected. Some options for improvements have been indicated in the uploaded paper with my comments.

The paper should improve the description of the various PCA analysis, especially with respect to the traits put into the PCA analysis. It is possible to show the coefficients of the different traits on the biplots. That would show the reader how the axes are made up.

Also, it should be investigated how an overall analysis with all accessions in one PCA can be done.

Lastly, the relationship between the biodiversity dendrograms and the PCA plots and the theme of the paper (sexual types) should be made more clear.

Experimental design

OK, but an overall PCA analysis could be welcome, plus a description of the make up of the PCA 1 and 2 axes in the overall analysis (still to be shown) or the individual PCA analysis.

Validity of the findings

In principle OK, but the authors should elaborate more on the theme of the sexual types in relation to biodiversity analysis.

Additional comments

See also all the comments in the uploaded commented version. Click on the comment balloons to read all of them. Parts on which comments have been given are in yellow.

·

Basic reporting

The manuscript reports good information and will helpful for the genetic improvement of the species via breeding.

Experimental design

The experiments conducted and analysis made were to the scientific standards

Validity of the findings

The finds reported were good standards and accepted to the scientific level

Additional comments

The present manuscript entitled "Sex expression and floral diversity in Jatropha curcas: A population study in its center of origin" The authors have put the best efforts to characterize the sex expression and floral diversity of potential species Jatropha curcus, The species is having significant importance as a biodiesel plant having good economic importance and because of the improper distribution by anthropogenic activity due to its economic importance the species is losing its valuable natural genetic resources and the research work conducted is very much relevant to the present context and much needed.
The over all the work conducted and analysis made is in good scientific standards and manuscript suitable for the publication. However to review these minor points given bellow:

Corrections:
• In title the name of the plant should include “L” correct it to “Jatropha curcas L”
• In few places of the manuscript the plant name is not in italics
 Figure 1, 2, 3, 4, 5, Table 1.

The Manuscript is discussed well and I recommend strongly for publication.

---

## Round 0.2 · accepted · Accept

The revised version of the manuscript can be accepted for publication.

·

Basic reporting

authors responded well to the comments

Experimental design

is good

Validity of the findings

is good

Additional comments

review was made accordigly